

# Effects of indium exposure on respiratory symptoms: a retrospective cohort study in Japanese workers using health checkup data

Toshiharu Mitsuhashi

Center for Innovative Clinical Medicine, Okayama University Hospital, Okayama, Japan

Corresponding author
Toshiharu Mitsuhashi,
mitsuh-t@cc.okayama-u.ac.jp

## ABSTRACT

**Background**. Indium compounds are known health hazards for lung cancer and interstitial pneumonia. Furthermore, they are related to emphysema, alveolar proteinosis, and cholesterol granuloma. In Japan, laws were revised in 2013 to tighten regulations on indium exposure in workplaces. However, its impact on the health of workers who handle indium has not been evaluated. This study aimed to investigate whether subjective respiratory symptoms in these workers have reduced after the 2013 amendment in the regulations.

**Methods**. The subjects were workers from certain areas of Japan who had undergone health checkups between January 1, 2013, and June 30, 2015. Indium-handling and non-handling workers were categorized into the exposed and less-exposed groups, respectively. Based on the findings of health checkups during this period, the hazard ratio of subjective respiratory symptoms (cough, sputum production, shortness of breath, and palpitation) and its 95% confidence intervals (CIs) were calculated with the less-exposed group as the reference. The Prentice-Williams-Peterson model was used for calculation, and a model that adjusted for coarse analysis and potential confounding factors was adopted.

**Results**. Overall, 2,561 workers (from 22 companies) who underwent 6,033 health checkups were included. The total person-years were 2,562.8 years, and 162 outcome events occurred. The hazard ratios of the exposed group were 1.65 (95% CI [1.14–2.39]: $p = 0.008$) and 1.61 (95% CI [1.04–2.50]: $p = 0.032$) in the crude and adjusted models, respectively.

**Conclusion**. Indium-handling workers had a high hazard of the subjective respiratory symptoms than non-indium -handling workers despite stricter regulations on indium exposure in workplaces. This indicates the need for further changes to the legislation to protect the health of workers exposed to harmful substances in workplaces. Further studies including larger diverse cohorts are needed to validate our findings.

## INTRODUCTION

Indium compounds are important materials in the manufacture of information devices. Indium tin oxide (ITO) thin films and indium phosphide semiconductors (In-P) are used for making electrodes of flat panel and liquid crystal displays owing to their transparency and good conductivity. The increase in the availability of information devices has increased the amounts of indium used. According to the statistics of the *Japan Oil Gas and Metals National Corporation, 2018*, Japan was the largest global consumer of indium mineral ore in 2017.

Until the mid-1990s, little was known regarding the toxicity of indium. The first case of interstitial pneumonia that was probably related to ITO inhalation was reported in 2003 (*Homma et al., 2003*). Indium compounds have subsequently been demonstrated to be associated with respiratory diseases such as interstitial pneumonia and lung cancer (*Chonan, Taguchi & Omae, 2007*; *Hamaguchi et al., 2008*; *Cummings et al., 2010*; *Cummings et al., 2012*; *Omae et al., 2011*; *Nakano et al., 2014*; *Nakano et al., 2015*; *Nakano et al., 2019*). Furthermore, they are related to emphysema, alveolar proteinosis and cholesterol granuloma (*Chonan et al., 2019*). The International Agency for Research on Cancer classified In-P and ITO as 2A (probably carcinogenic to humans) and as 2B (possibly carcinogenic to humans) carcinogens, respectively (*IARC Working Group on the Evaluation of Carcinogenic Risks to Humans, 2006*; *IARC Working Group on the Evaluation of Carcinogenic Risks to Humans, 2018*).

Reports also suggest that indium-handling workers, who are constantly engaged in the production and/or handling of indium, often have respiratory symptoms. In a cross-sectional study that was conducted prior to tightening of the Japanese regulations, the prevalence of subjective respiratory symptoms among indium smelting and soldering workers was 6.1% and 8.1%, respectively (*Nakano et al., 2015*). As indium usage will increase in the future, the prevalence of respiratory symptoms is also expected to increase. Therefore, the adverse impact of increasing exposure to indium on the respiratory system is an important health issue.

Under these circumstances, the Ministry of Health, Labor and Welfare (MHLW) in Japan established the prevention guidelines for indium-handling workers in 2010 (*Ministry of Health Labor and Welfare, 2010*). In 2013, The MHLW added indium to the list of substances regulated by the Ordinance on Prevention of Hazards due to Specified Chemical Substances (*Ministry of Health Labor and Welfare, 2013*). As per the ordinance, companies were obligated to install equipment to control the diffusion of indium into the air and to measure the concentration of indium dust in the air every six months. Further, in addition to general health checkups, employers are obligated to perform special medical examinations (indium medical examination) for indium-handling workers and provide employees with education on indium-related health hazards.

The tightening of regulations in Japan was expected to reduce the severe consequences of long-term and high-concentration indium exposure. However, some workers continue to present with subjective complaints of respiratory symptoms, such as cough and sputum

production. The association between the onset of subjective respiratory symptoms and the strengthening of regulations in Japan has not been evaluated.

Thus, the present study aimed to investigate whether the prevalence of subjective respiratory symptoms in Japanese indium-handling workers has been reduced after tightening of the regulations. Health checkup data were used to examine the prevalence of subjective symptoms in indium-handling workers.

## MATERIALS & METHODS

### Study design and setting

This was a retrospective cohort study that used data obtained during a health check by the Chugoku Occupational Health Association between January 1, 2013 and June 30, 2015. In Japan, the Industrial Safety and Health Act requires employers to ensure that workers receive health checkups at least annually. Because this study used anonymized health checkup data, details about the companies and individuals were not available.

### Subjects

The subjects were workers in indium-handling companies from Japan who had undergone a health checkup at the Chugoku Occupational Health Association during the study period. Subjects who had a history of respiratory disease or who were receiving treatment for the same, those with deficient outcomes, and those with outcomes from the start point were excluded.

The data of subjects who met all and none of the inclusion and exclusion criteria, respectively, were used in the final analysis.

In this study, new occurrence of respiratory symptoms was defined as outcomes, and thus observations in workers who already had outcome were excluded. Therefore, only the population at risk was included in the analysis.

### Description of the dataset

The characteristics of the obtained data are shown in Fig. 1. The case where worker $i$ received $m$ health checkups has been illustrated as an example. The health checkups were nested among workers as per the data structure (Fig. 1). In the analysis, the period from the $(k-1)th$ health checkup to the $k$ th health checkup was treated as one observation. Exposure variables and covariates were obtained at the $(k-1)th$ health checkup, and outcome variables were obtained at the $kth$ health checkup. At $k = 1$, the observation start date was set to April 1 (first health checkup was from April to September) or October 1 (first health checkup was from October to March). This was done as in Japan, employment and relocation often starts on these days. Therefore, changes in employment and relocation related to the exposure status often occur from these days. In addition, the observations for all subjects were censored on the last health checkup before June 30, 2015.

### Data collection
#### Exposure variable

The workers who underwent health checkups and indium medical examinations simultaneously at the start of the observation ($k-1th$ health checkup) period were defined

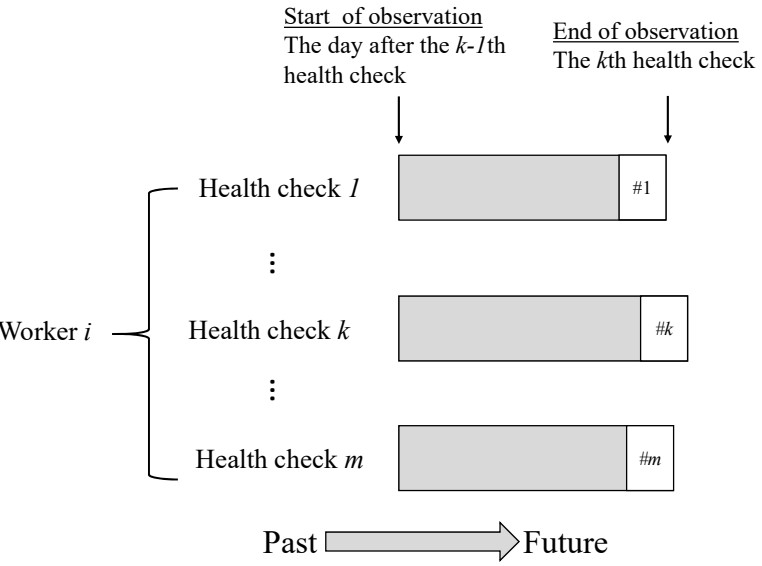

**Figure 1 Data structure of the study sample.**

as exposed. In Japan, the indium medical examination is mandatory at the time of hiring or reassignment to work and once every six months thereafter. Therefore, those who underwent indium medical examinations within six months before and after the observation start point were also considered exposed. This means that undergoing an indium medical examination is consistent with indium handling experience. Thus, it is reasonable to define this as being exposed. However, because this study used anonymized health checkup data, the details of the work (refining, sintering, vapor deposition) were unknown.

Non-indium-handling workers of indium-handling companies were defined as less-exposed rather than non-exposed because they may have some form of indium exposure such as through air in the company.

### Outcome variable

The outcome variable was the new occurrence of respiratory symptoms and was evaluated at the end of the observation period (*kth* health checkup). This was defined as positive when the worker answered "yes" to one or more of the following four items on the health checkup questionnaire: (1) Does your voice wither?, (2) Do you have a foreign body sensation in your throat?, (3) Has your coughing continued or is there any sputum production?, and (4) Do you have strong palpitations or breathlessness?

### Potential confounders and variables about exposed workers

As potential confounding factors, data on age, gender, smoking status, and current and past work experience in a dusty environment were obtained from the *k-1th* health checkup.

In addition, we collected data on the proportion of indium-handling workers who wore protective equipment and on the serum concentrations of indium (S-In) and Krebs von

den Lungen-6 (KL-6). Non-indium-handling workers have not received indium medical examination, and thus there were no data for protective equipment, S-In, and KL-6.

### Efforts to address potential sources of bias

The missing and incorrect values (i.e., cases where a character string was input for a continuous variable, among others) were reconfirmed with the staff of the Chugoku Occupational Health Association, and correct values were obtained as far as practicable.

## Statistical analysis

Observations with missing values for one or more of the variables required for analysis were excluded. All statistical analysis were performed using the Stata (Stata Corporation, version 15.1, College Station, TX, USA) software package. All $p$ values were two-sided, and those less than 0.05 were considered to be statistically significant.

### Descriptive statistics

Descriptive statistics of the first health checkup data of each worker were calculated.

### Inferential statistics

The first event occurring for each indium exposure during the first medical checkup was graphed as the Nelson-Aalen cumulative hazard estimate. The effects of indium exposure on complaints of respiratory symptoms were assessed using the hazard ratio (HR). After verifying the proportional hazards assumption, the Cox proportional hazards model was used to calculate the HR of the effect of exposure on the first event occurrence.

However, complaints of respiratory symptoms may occur in multiple instances in one subject during the observation period. Therefore, multiple failure events may be noted. Additionally, the state of indium exposure may change during the observation period. Therefore, the HRs and their 95% confidence intervals were calculated for all event occurrences using the Prentice-Williams-Peterson (PWP) model (*Prentice, Williams & Peterson, 1981*).

The crude and adjustment models were used for individual analysis. In the adjustment model, all potential confounders were used as covariates.

### Sensitivity analysis

Because the exposed workers may have reported subjective respiratory symptoms more frequently than those who were less exposed, the HR may be overestimated. The sensitivity analysis was necessary based on the subjective symptoms that were considered to be biologically unrelated to indium exposure. In cases where the HR of subjective respiratory symptoms was significantly higher exclusively due to overreporting, the sensitivity analysis would also provide a significantly higher HR for the same reason.

In July 2019, several databases including Medline, PubMed, Cochrane Library, and Google Scholar were searched for reports regarding the relationship between indium exposure and sleep disorders; no reports suggesting any relationship were found. Therefore, the sensitivity was analyzed in a similar manner to the main analysis, using sleep disorder as an outcome variable. On sensitivity analysis, those who reported sleep disorders at the start of observation or had a history of mental illness were excluded.

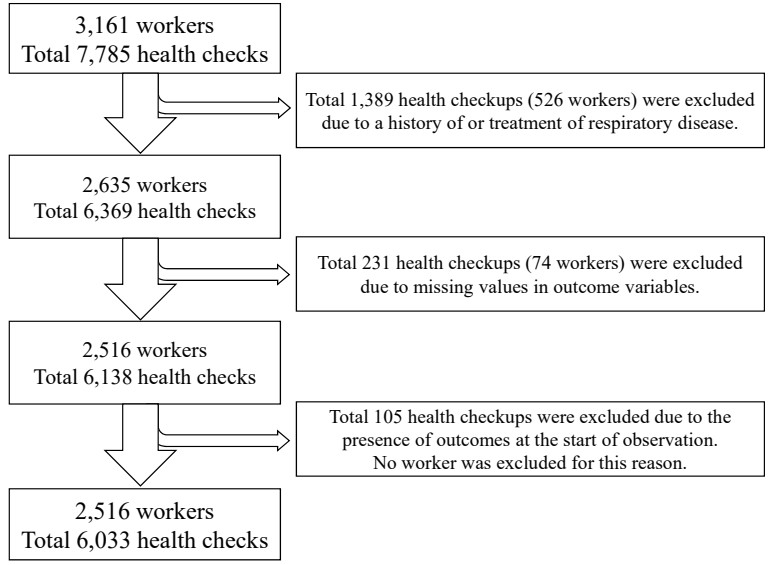

**Figure 2    Flow of workers and observations in the study.**

## Ethical issues

This study was approved by the Ethics Committee of the Okayama University Graduate School of Medicine, Dentistry and Pharmaceutical Sciences and Okayama University Hospital (approval number K1607-008).

Because of the retrospective nature of the study, the ethics committee waived the need for obtaining informed consent from patients. The web page provides adequate information on the purpose and methods of the study for the benefit of the potential subjects. The web page also mentions how to decline participation and states that subjects are free to decline participation for any reason.

## RESULTS

### Descriptive statistics

The data of 3,161 workers from 22 companies were analyzed. The health checkups (i.e., observations) were performed for a total of 7,758 times. The worker selection and observation flowchart are summarized in Fig. 2. In cases where all health checks nested for a particular worker were excluded, the worker was also excluded (Fig. 2).

Initially, 1,389 observations and 526 workers were excluded owing to the presence of a history of treatment for respiratory disease; 231 observations with missing outcomes were then excluded along with 74 workers. These exclusions were necessary as 105 observations had an outcome at the start of the observation (*k-1th* health checkup). Finally, data of 6,033 observations from 2,561 workers of 22 companies were analyzed. The total person-years were 2,562.8 years, and a total of 162 outcome events occurred.

The demographic characteristics of the exposed and less-exposed workers at the first health checkup are shown in Table 1. At the first health checkup, 412 and 2,149 workers were

**Table 1 Descriptive characteristics of the 2,561 workers at the first follow-up health check according to indium exposure.**

| | | Indium exposure | |
| --- | --- | --- | --- |
| | | Exposed group (n = 412) | Less-exposed group (n = 2,149) |
| Sex | Male | 347 (84.2%) | 1746 (81.2%) |
| | Female | 65 (6.8%) | 403 (7.9%) |
| Age (years) | Mean ± SD | 35.24 ± 7.64 | 35.79 ± 8.45 |
| Smoking | Non-smoker | 175 (42.5%) | 1055 (49.1%) |
| | Former smoker | 57 (13.8%) | 298 (13.9%) |
| | Current smoker | 180 (43.7%) | 796 (37.0%) |
| Current work experience in a dusty environment | | 49 (11.9%) | 40 (1.9%) |
| Past work experience in a dusty environment | | 51 (12.4%) | 13 (0.6%) |

Notes.
SD, standard deviation

exposed and less exposed, respectively. There were no large differences between the groups in terms of sex, age, and smoking status. However, there were large differences between groups with current and past work experience in a dusty environment. In the exposed group, 49 (11.9%) workers were currently working in a dusty environment, whereas 40 (1.9%) were in the less-exposed group. Among those with past working experience in a dusty environment, 51 (12.4%) and 13 (0.6%) were in the exposed and less-exposed groups, respectively (Table 1).

Table 2 shows the number of protective equipment wearers, the number of experienced workers at high exposure, chest radiograph findings, and S-In and KL-6 levels in the exposed group at the first health checkup. In total, 432 workers had been exposed at least once during the observation period. Overall, 414 (95.8%) workers always wore protective equipment, and none had experienced heavy exposures. The chest radiographs showed no abnormal findings in 400 (92.6%) workers. The S-In and KL-6 values were low, and their means and standard deviations were $0.15 \pm 026$ µg/L and $231.59 \pm 88.48$ U/mL, respectively; the standard values (less than 3 µg/L for S-In and less than 500 U/mL for KL-6) were exceeded in nine and one workers, respectively. Among them, the standard values of both serum markers had been exceeded in one worker (Table 2).

Figure 3 shows the cumulative incidence of the first complaint of respiratory symptoms; this was high in the exposed group. However, the log-rank test did not demonstrate statistical significance ($p = 0.052$).

## Inferential statistics

The upper half of Table 3 shows the results of the Cox proportional hazard model considering only the first event during the observation period. The hazard ratios of the exposed group, with reference to the less-exposed group, were 1.42 (95% CI [0.99–2.04]) and 1.50 (95% CI [1.04–2.17]) in the crude and adjusted models, respectively.

**Table 2  Descriptive characteristics of 432 workers exposed to Indium compound more than once.**

| | | |
|---|---|---|
| Wears dust respirator | Always | 414 (95.8%) |
| | Sometimes | 3 (0.7%) |
| | Not | 11 (2.5%) |
| | No answer | 4 (0.9%) |
| Experience of massive exposure | Yes | 0 (0%) |
| | No | 427 (98.8%) |
| | Unknown | 5 (1.2%) |
| Chest radiograph findings | No findings | 400 (92.6%) |
| | Pleural adhesions | 7 (1.6%) |
| | Sclerotic lesions | 5 (1.2%) |
| | Others | 20 (4.6%) |
| Serum indium ($\mu$g/L) | Mean $\pm$ SD | 0.15 $\pm$ 0.26 |
| Exceeded standard value (3 $\mu$g/L) | | 1 (0.2%) |
| KL-6 (U/mL) | Mean $\pm$ SD | 231.59 $\pm$ 88.48 |
| Exceeded standard value (500 U/mL) | | 9 (2.1%) |

**Notes.**
SD, standard deviation

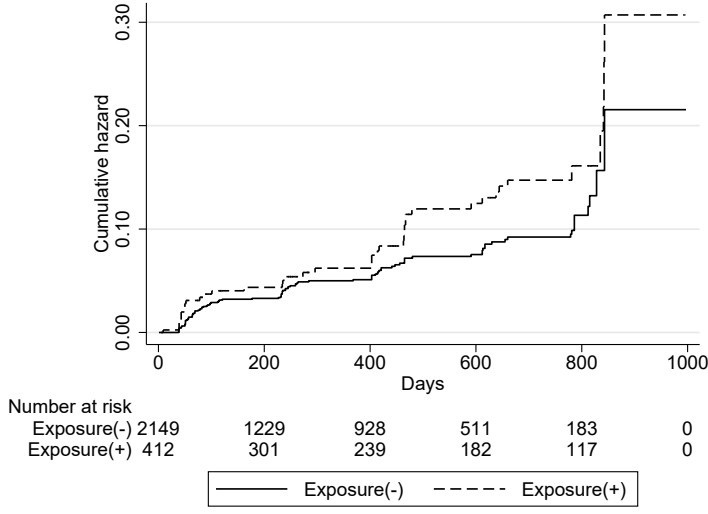

Number at risk

| | | | | | |
|---|---|---|---|---|---|
| Exposure(-) | 2149 | 1229 | 928 | 511 | 183 | 0 |
| Exposure(+) | 412 | 301 | 239 | 182 | 117 | 0 |

**Figure 3  Cumulative hazard of the first complaint of respiratory symptoms.**

The results of the PWP model are shown in the lower half of Table 3. The hazard ratios in the crude and adjusted models were 1.65 (95% CI [1.14–2.39]) and 1.61 (95% CI [1.04–2.50]), respectively.

## Sensitivity analysis

The results of sensitivity analysis are shown in Table 4. For complaints of sleep disorders, the hazard ratios in the crude and adjusted models were 1.11 (95% CI [0.78–1.58]) and 1.31 (95% CI [0.86–1.98]), respectively. There were no statistically significant differences in the hazard ratios between the exposed and the less-exposed groups.

**Table 3  Crude and adjusted Cox and PWP model-based predictions of complaints of respiratory symptoms among 2,561 workers.**

|  | Crude model | | Adjusted model | |
| --- | --- | --- | --- | --- |
|  | HR (95% CI) | *p*-value | HR (95% CI) | *p*-value |
| **Cox model** | | | | |
| Indium exposure | 1.42 (0.99, 2.04) | 0.056 | 1.50 (1.04, 2.17) | 0.032 |
| Male | | | 1.00 (reference) | – |
| Female | | | 1.62 (1.05, 2.50) | 0.029 |
| Age (per 5 years) | | | 0.88 (0.80, 0.97) | 0.008 |
| Non-smoker | | | 1.00 (reference) | – |
| Former smoker | | | 1.15 (0.67, 1.97) | 0.616 |
| Current smoker | | | 1.76 (1.24, 2.51) | 0.002 |
| Current work experience in a dusty environment | | | 0.89 (0.44, 1.81) | 0.755 |
| Past work experience in a dusty environment | | | 0.61 (0.26, 1.43) | 0.258 |
| **PWP model** | | | | |
| Indium exposure | 1.65 (1.14, 2.39) | 0.008 | 1.61 (1.04, 2.50) | 0.032 |
| Male | | | 1.00 (reference) | – |
| Female | | | 1.60 (1.04, 2.46) | 0.031 |
| Age (per 5 years) | | | 0.87 (0.79, 0.95) | 0.003 |
| Non-smoker | | | 1.00 (reference) | – |
| Former smoker | | | 1.19 (0.72, 1.99) | 0.495 |
| Current smoker | | | 1.72 (1.22, 2.42) | 0.002 |
| Current work experience in a dusty environment | | | 0.99 (0.57, 1.73) | 0.968 |
| Past work experience in a dusty environment | | | 1.21 (0.66, 2.24) | 0.535 |

**Notes.**
CI, Confidence Interval

**Table 4  Crude and adjusted PWP model-based predictions of complaints regarding sleep disorders among 3,086 workers.**

|  | Crude model | | Adjusted model | |
| --- | --- | --- | --- | --- |
|  | HR (95% CI) | *p*-value | HR (95% CI) | *p*-value |
| Indium exposure | 1.11 (0.78, 1.58) | 0.544 | 1.31 (0.86, 1.98) | 0.202 |
| Male | | | 1.00 (reference) | – |
| Female | | | 1.49 (1.07, 2.09) | 0.019 |
| Age (per 5 years) | | | 0.95 (0.88, 1.03) | 0.240 |
| Non-smoker | | | 1.00 (reference) | – |
| Former smoker | | | 1.01 (0.68, 1.50) | 0.958 |
| Current smoker | | | 1.02 (0.77, 1.35) | 0.897 |
| Current work experience in a dusty environment | | | 0.89 (0.55, 1.43) | 0.616 |
| Past work experience in a dusty environment | | | 0.82 (0.48, 1.41) | 0.476 |

**Notes.**
CI, Confidence Interval

## DISCUSSION

In this study, there was a significant large risk of subjective respiratory symptoms in the indium-handling workers than non-indium-handling workers despite strengthened Japanese regulations. These results may suggest that regulatory strengthening did not reduce the prevalence of subjective respiratory symptoms among indium-handling workers.

There are three explanations for these results. First, the subjective respiratory symptoms may have been caused by cumulative exposure. A previous study reported that higher cumulative exposures were associated with a higher incidence of dyspnea, lower spirometry parameters, and higher serum biomarkers of lung disease (*Cummings et al., 2016*). It is possible that the indium-handling workers in this cohort had more symptoms owing to the cumulative exposure prior to the tightening of regulations. Second, due to the long biological half-life of indium (8.09 years; *Amata et al., 2015*), pre-regulatory exposure effects may remain in the 3-year observation period. For this reason, research with longer observation period is necessary. The third factor is believed to be the use of protective equipment. In this study, 95.8% of the indium-handling workers were always wearing the dust respirator. This was not considered a high proportion, as 100% of workers are expected to wear protective equipment in a dusty environment, as per the Japanese law. In addition, reports suggest inappropriate working conditions in small companies (*Aiba et al., 1995*). According to this report, 73.1% of workers were wearing the dust respirator inappropriately, while the respirators were unauthorized in 22%. Furthermore, although reports suggest that powered air-purifying respirators (PAPRs) have a higher removal capacity (*Liu et al., 2016*), Japanese regulations do not mandate the wearing of PAPR in workplaces with low indium concentrations. Therefore, the indium exposure to the respiratory tract was not reduced despite tightening of the regulations, and the prevalence of subjective symptoms was higher than that in the less-exposed workers.

The HR was higher in the PWP than that in the Cox model. Because only the first symptom was analyzed in the Cox model, the symptom occurrences in the subsequent observation period were ignored. The HR was underestimated by the Cox model. Therefore, the HR provided by the PWP model, which analyzed all observation periods, reflected the true value better than the Cox model.

In cases where the exposed workers had over-reported any subjective symptoms, the sensitivity analysis would have demonstrated significant results, but the sensitivity analysis did not yield significant results. Therefore, the impact of over-reporting was considered to be minimal.

Although there were significant differences in subjective respiratory symptoms in this cohort, the increase in biomarkers was minimal. Only nine (2.1%) workers had S-In and KL-6 levels that exceeded the standard values. This is in contrast to results from previous studies that reported an increased risk of lung cancer and interstitial pneumonia with indium exposure. In an 11-year cohort study (*Nakano et al., 2019*), the S-In and KL-6 levels were higher in indium-handling workers than in those who were not exposed. Other studies have also reported higher S-In, KL-6, and SP-D levels among indium-handling

workers (*Liu et al., 2012*; *Choi et al., 2013*). This difference may be attributable to reductions in exposure to a certain extent, consequent to the tightening of regulations.

Female had significantly higher HR for respiratory symptoms. However, in the sensitivity analysis, the HR was also significantly higher, which may be due to the effect of over-reporting.

### Strengths and limitations

The strengths of this research are as follows. First, the impact of random errors was extremely small owing to the large sample size. Second, this study employed the PWP model; as this model incorporates the probability of multiple failures into the statistical model, it is more realistic, and is likely to provide a more accurate estimate. Third, the indium medical examination was used to assess indium exposure. As this examination is mandated by Japanese law, misclassification of exposure was minimized.

However, there are also certain limitations. First, misclassifications may have resulted owing to the subjective outcomes. The exposed group was more likely to respond affirmatively to items "with symptoms" on the questionnaire. In cases where this tendency was strong, the HR was likely to be higher than the true value. However, sensitivity analysis did not demonstrate significant HRs. Therefore, the high HR of subjective respiratory symptoms may not be solely attributable to the influence of the response tendency of the exposed group. Second, the details of the exposure were unknown. In this study, the indium concentrations and nature of work were unknown, and they may have been large variations in exposure. This may have distorted the HR towards a value of one. However, this influence was likely to be minimal owing to the levels of significance. Third, exposures and outcomes were only evaluated during the health checkup, and data acquisition was delayed as the true timing of the exposures and outcomes was earlier than that of the health checkup. However, because this delay was similar for exposures and outcomes, it was likely to have no or minimal impact on the analysis. Fourth, the exposure group reported a higher proportion of dust work environment, which may confound the relationship between exposure and may have led to overestimations of the HRs. Meanwhile, because this confounding factor was adjusted using a statistical model, overestimation is minimal. Fifth, in this study, more sensitive biological markers such as S-In could not be compared between the exposed and the less-exposed groups. Therefore, it was not possible to conduct a more detailed analysis. However, S-In and KL-6 were at low levels even in the exposed group, and thus the effect was considered to be small.

### Generalization of the study results

Because only companies and workers from one region of Japan were included in this study, the scope of the results may be limited by the geographical characteristics. Therefore, these results should be generalized only after careful consideration.

## CONCLUSIONS

Although there were no elevations in the levels of biomarkers, the hazard ratio of subjective respiratory symptoms was significantly higher in the indium-exposed group than in the

less-exposed group, even after the regulations in Japan were tightened. This indicates the need for further changes to the legislation to protect the health of workers exposed to harmful substances in workplaces. This includes strengthening of health management and increasing the use of more effective protective equipment such as PAPRs, among others. Further studies including larger diverse cohorts with a more valid questionnaire and biomarker are needed to validate our findings.

## ACKNOWLEDGEMENTS

The author is grateful to the Chugoku Occupational Health Association for providing health checkup data. The author would also like to thank Editage for English language editing.

### Funding

This study was supported by the Chugoku Occupational Health Association. The funders had no role in study design, data collection and analysis, decision to publish, or preparation of the manuscript.

### Grant Disclosures

The following grant information was disclosed by the author:
Chugoku Occupational Health Association.

### Competing Interests

The author declares there are no competing interests.

### Author Contributions

- Toshiharu Mitsuhashi conceived and designed the experiments, performed the experiments, analyzed the data, prepared figures and/or tables, authored or reviewed drafts of the paper, and approved the final draft.

### Human Ethics

The following information was supplied relating to ethical approvals (i.e., approving body and any reference numbers):

The study was approved by the Okayama University Graduate School of Medicine, Dentistry and Pharmaceutical Sciences and Okayama University Hospital, Ethics Committee (approval number K1607-008).

### Data Availability

The raw data and code are available in the Supplementary Files.

### Supplemental Information

Supplemental information for this article can be found online at http://dx.doi.org/10.7717/peerj.8413#supplemental-information.

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
