# Peer review of "Effects of indium exposure on respiratory symptoms: a retrospective cohort study in Japanese workers using health checkup data"

_PeerJ, doi:10.7717/peerj.8413_

## Round 0.1 · original submission · Major Revisions

Please consider the comments of the reviewers and modify your manuscript accordingly. Please pay particular attention to concerns that perhaps some conclusions need to be softened based on the limited data supporting them.

Reviewer 1 ·

Basic reporting

No comment

Experimental design

No comment

Validity of the findings

No comment

Additional comments

To the Authors:
The authors conduct a retrospective cohort study of workers from 22 companies in one region of Japan that handle some form of indium, which seems to automatically enroll workers from these companies in a form of medical surveillance required by Japanese law that includes at least annual health checkup or ‘indium medical checkups’ every six months based on likelihood of exposure that seems to include blood draws for serum indium and KL-6 monitoring. Workers were considered exposed if they ‘handle indium’ and unexposed if they do not ‘handle indium.’ The outcome of interest is self-reported new respiratory symptoms during a health checkup. The authors calculate Hazard Ratios using several modeling approaches to conclude indium-exposed workers are more likely to develop new respiratory symptoms during the study compared with unexposed workers, despite very low serum indium and KL-6. Study design and statistical methods were applied to reduce potential biases, which strengthens conclusions.
Some of the conclusions might benefit from more explanation or clarification. For example, the success of the 2013 regulatory changes are questioned, based on the biomarkers presented, it would seem the regulations/prevention strategies are largely successful and should be celebrated. Perhaps, there are aspects that could be improved, but more specifics might help clarify. Also, the authors suggest the prevalence of symptoms in indium-exposed workers has increased or at least ‘were not reduced,’ and there don’t seem to be any prevalence or incidence data presented to draw this conclusion. I also wonder if the dusty work environments reported by a higher proportion of indium-exposed workers should be included in the limitations. I know the modelling included this as a covariate, but perhaps you can clarify how the new respiratory symptoms can be attributed to indium rather than dust. Lastly, it would be useful to include additional information on what is considered ‘indium handling’ as it defines the exposure and is not necessarily intuitive. I hope you find these general comments useful and provide a few more specific comments below, but commend you on a well designed study and well written manuscript.
Abstract: Line 23: Does non-handling = non-exposed? Less-exposed perhaps, but unlikely non-exposed.
Line 33: ‘respiratory symptoms…were not reduced’ Can you conclude this? There is no comparison to pre-2013 regulations to make this comparison.
Introduction: Line 45: Does global consumption equal production?
Line 55: Define ‘indium-handling workers’
Line 57: symptom > symptoms
Line 58-59: Last sentence of paragraph is confusing and should be reworked.
Line 67: Expand on ‘indium medical examination’ – what it involves and who gets it and why? Lung function testing included?
Line 75: have reduced > have been reduced
Materials and Methods: Line 91: any more information on the range of ‘indium-handling companies?’
Line 124: Can voice wither be attributed to indium?
Line 126: Is a palpitation a respiratory symptom?
Line 132: Define ‘KL-6.’
Results: Line 196: Consistent use of commas in numbers, ie 6,033 – likely based on journal specifications
Line 202+: How can respiratory symptoms be attributed to indium when indium-exposed workers were more likely to work in dusty environments?
Line 212: These are very low biomarkers and that should be discussed at some point in manuscript.
Discussion: Line 237: ‘regulatory strengthening did not reduce the prevalence of subjective respiratory symptoms’ seems too strong of a conclusion based on findings presented in manuscript.
Line 264: Define ‘standard values.’
Conclusions: Line 301: no pre-regulations/2013 data presented, so symptom prevalence cannot be tied to regulations?
Figure 1: is there a component of time from left to right that should be labeled?
Figure 3: Label Y axis.
Table 1: Statistical test and P values for demographic comparison of exposed and non-exposed? Also, probably shouldn’t be considered non-exposed, perhaps less exposed?
Tables 3 and 4: Females more likely to report respiratory symptom, is this expected? Should it be discussed in manuscript?

Reviewer 2 ·

Basic reporting

no comments

Experimental design

The author analyzed using data from three years, however, the onset of worker's respiratory symptoms should be analyzed the data with the work conditions or the situation of exposure through the questionnaire at the annual special health check ups. The persuasiveness of the whole paper was reduced in that respect because the author analyzed only using the data of the presence or absence of respiratory symptom symptoms as output.

Validity of the findings

Scientific evidence can be found in the epidemiological calculations, however, it seems that the rationale is not sufficient to conclude that strengthening legislation did not reduce the incidence of respiratory symptoms in indium workers.
The author concludes that the use of protective equipment should be strengthen, however according to the results of this paper, 92.6% of indium workers are using protective equipment. This is a contradiction, and it will be necessary to discussed more carefully.

Additional comments

This is an interesting paper on the development of respiratory symptoms in indium workers.
However, there are points that should be revisited overall.

Reviewer 3 ·

Basic reporting

no comment

Experimental design

Indium compounds, especially indium-tin oxide, are used to make liquid crystal displays and have recently been recognized to evoke pulmonary toxicity when inhaled as respirable particles. Far east Asia including Japan produces the major part of the indium compounds; in 2010 the Japanese MHLW issued a technical guideline for preventing health hazards of ITO workers and in 2013 it was enacted. This paper compared the subjective respiratory symptoms in Japanese indium workers before and after the legislation; the author measured HR of respiratory symptoms scores of indium workers relative to that of non-indium workers retrospectively. HR did not decrease after the legislation and the author concluded that indium workers’ symptoms did not decrease after the introduction of strict regulations. The aim of the study is important and clearly stated, however, there is a few questions that remains unanswered.

Validity of the findings

no comment

Additional comments

Major Comments

1. It has been reported that respired indium stays for a long time in the body including the lung; the biological half -life of the serum indium(sIn) is estimated to be 8.09 years (Thorax, 2015,V70, pp 1040), therefore, two to three years in between the samplings cannot separate the exposure levels sharply. Longer intervals would be better. Alternately, how about focusing the two groups of newly engaged workers naïve to indium processing, before and after the new rules. That would give us clearer data. Would such comparison available in your data?

2. Symptomatic changes are important; however, they may not be a sensitive marker to detect the effects of tightening regulations. According to the guideline of MHLW, employers are required to measure the workers’ sIn every six months, and it has been reported that sIn is correlated with the degree of radiological (interstitial or emphysematous changes), functional (diffusing capacity of the lung for carbon monoxide) and serum biomarkers of interstitial pneumonia (KL-6 and SP-D), therefore, sIn, rather than symptoms, appears to be a sensitive marker in assessing the effects of implementing new regulations.

Minor Comments

L16 (Abstract)
I think the author should add emphysema, alveolar proteinosis and cholesterol granuloma as they are characteristic, although not specific, findings of indium lung (Tohoku J. Exp. Med. 2019, V248, pp143).
L94 exclusion criteria
It appears that the exclusion criteria adopted in this study could omit some heavily exposed workers, who had engaged in indium processing for long durations.

---

## Round 0.2 · accepted · Accept

Thank you for your efforts in revising your manuscript according to reviewer comments.

Reviewer 1 ·

Basic reporting

No comment

Experimental design

No comment

Validity of the findings

No comment

Additional comments

The manuscript is much improved following revisions and I commend the author for his efforts. The manuscript warrants publication and contributes meaningfully to the field.